# The Role of Vitamin K in Humans: Implication in Aging and Age-Associated Diseases

**DOI:** 10.3390/antiox10040566

**Published:** 2021-04-06

**Authors:** Daniela-Saveta Popa, Galya Bigman, Marius Emil Rusu

**Affiliations:** 1Department of Toxicology, Faculty of Pharmacy, Iuliu Hatieganu University of Medicine and Pharmacy, 8 Victor Babes, 400012 Cluj-Napoca, Romania; 2The Baltimore Geriatric Research, Education and Clinical Center, Veterans Affairs Maryland Health Care System, Baltimore, MD 21201, USA; galya.bigman@va.gov; 3Department of Pharmaceutical Technology and Biopharmaceutics, Faculty of Pharmacy, Iuliu Hatieganu University of Medicine and Pharmacy, 8 Victor Babes, 400012 Cluj-Napoca, Romania; rusu.marius@umfcluj.ro

**Keywords:** vitamin K, phylloquinone, menaquinone, menadione, osteocalcin, matrix Gla protein, bone health, COVID-19, osteoporosis, vascular calcification

## Abstract

As human life expectancy is rising, the incidence of age-associated diseases will also increase. Scientific evidence has revealed that healthy diets, including good fats, vitamins, minerals, or polyphenolics, could have antioxidant and anti-inflammatory activities, with antiaging effects. Recent studies demonstrated that vitamin K is a vital cofactor in activating several proteins, which act against age-related syndromes. Thus, vitamin K can carboxylate osteocalcin (a protein capable of transporting and fixing calcium in bone), activate matrix Gla protein (an inhibitor of vascular calcification and cardiovascular events) and carboxylate Gas6 protein (involved in brain physiology and a cognitive decline and neurodegenerative disease inhibitor). By improving insulin sensitivity, vitamin K lowers diabetes risk. It also exerts antiproliferative, proapoptotic, autophagic effects and has been associated with a reduced risk of cancer. Recent research shows that protein S, another vitamin K-dependent protein, can prevent the cytokine storm observed in COVID-19 cases. The reduced activation of protein S due to the pneumonia-induced vitamin K depletion was correlated with higher thrombogenicity and possibly fatal outcomes in COVID-19 patients. Our review aimed to present the latest scientific evidence about vitamin K and its role in preventing age-associated diseases and/or improving the effectiveness of medical treatments in mature adults ˃50 years old.

## 1. Introduction

Aging is a multifactorial process that gradually deteriorates the physiological functions of various organs, including the brain, musculoskeletal, cardiovascular, metabolic, and immune system leading to numerous pathological conditions with high rates of morbidity and mortality. Oxidative stress (OS) and chronic inflammation are fundamental pathophysiological mechanisms in the aging progression [1,2,3].

As human life expectancy is rising, age-related diseases will increase as well. Recent studies validated the importance of modifiable lifestyle factors, diet included, in the attenuation of pathological changes in mature adults [4]. Healthy fats, vitamins, minerals, polyphenolics, with antioxidant and anti-inflammatory activity, can increase the quality of life and influence the aging process, and among these factors, vitamin K (VK) has an important part [5].

VK is known for its role in synthesizing some blood-clotting proteins (K for koagulation in German). VK represents a fat-soluble family of compounds with a common chemical structure, a 2-methyl-1,4-naphthoquinone ring and a variable aliphatic side-chain. The variable aliphatic chain differentiates two isoforms: vitamin K1 (VK1) or phylloquinone (PK) and vitamin K2 (VK2), usually designated as menaquinone (MK). MK exists in multiple structures, which are distinguished by the number of isoprenyl units and saturation in the side-chain (MK-n, where n is the number of isoprenyl units) [6]. These acronyms were used interchangeably throughout this article. The most common subtypes in humans are the short-chain MK-4, which is the only MK produced by systemic conversion of phylloquinone to menaquinone, and MK-7 through MK-10, which are synthesized by bacteria. VK3 (menadione), without side-chain and classified as a pro-vitamin, is a synthetic form of this vitamin (Figure 1).

Dark green leafy vegetables are the main sources for dietary PK (e.g., collards, turnip, broccoli, spinach, kale), 70–700 μg/100 g, as well as several fruits (e.g., dried prunes, kiwifruit, avocado, blueberries, blackberries, grapes), 15–70 μg/100 g, and some nuts (pine nuts, cashews, pistachios), 10–75 μg/100 g [7,8]. In contrast, the main sources of VK2 are fermented foods, cheeses, eggs, and meats (Table 1) [9,10].

Although dietary PK in vegetables is the major source of the VK intake (80–90%), only 5–10% is absorbed, whereas MKs from dairy products are almost completely absorbed. PK, tightly bound to plant chloroplasts, as well as PK digested with some phytochemicals (e.g., saponins, tannins, fibers, phytates) found in pulses, is less bioavailable to human. Though, PK from collards and broccoli is more bioavailable than PK from spinach [11,12].

Both VK1 and VK2 are recognized as cofactors for enzyme γ-glutamyl carboxylase (GGCX), which converts glutamic acid (Glu) to a new amino acid γ-carboxyglutamic acid (Gla), in VK-dependent proteins (VKDPs) during their biosynthesis [13]. These VKDPs require carboxylation to become biologically active, and the negatively charged γ-carboxyglutamic acid residues have a high affinity for positively charged calcium ions [14].

VKDPs can be classified as hepatic and extrahepatic. The hepatic VKDPs are largely involved in blood coagulation. Extrahepatic VKDPs perform different tasks: osteocalcin (OC) regulates the bone formation and mineralization, the matrix Gla protein (MGP) is a potent inhibitor of vascular calcification, nephrocalcin is involved in kidney functions, the growth arrest-specific protein 6 (Gas6) in the development and differentiation of nervous system [15]. Additionally, some extrahepatic VKDPs (proteins C and S) inhibit coagulation by inactivating specific coagulation factors necessary to form blood clots [16].

Recent findings revealed the novel role of VK as an antioxidant and implicitly anti-inflammatory agent independent of its GGCX cofactor activity [17]. The antioxidant properties of VK are based on a protective action against oxidative cellular damage and cell death by (1) direct reactive oxygen species (ROS) uptake [17]; (2) the limiting of free radical intracellular accumulation [18], and (3) inhibition of the activation of 12-lipoxygenase [19].

Scientific evidence suggests that VK also has anti-inflammatory activity, a vital component against various chronic aging diseases [20]. VK inhibits the activation of the nuclear factor kappa B (NF-кB) and thus decreases the production of proinflammatory cytokines [17]. VK is significantly and inversely related to individual inflammatory biomarkers and inflammatory processes due to its anti-inflammatory effects [21].

The daily reference intake of VK is based mostly on bleeding-associated studies, and it varies between countries. US dietary guidelines recommend daily intakes of 90 and 120 μg for women and men, respectively, while the guidelines in the United Kingdom are set at 1 μg/kg body weight/day [11]. However, these recommendations are insufficient to induce complete carboxylation of all VKDPs. Only MK-7, having higher bioavailability and longer half-life, proved to promote γ-carboxylation of extrahepatic VKDPs at the current recommended levels, while the recommended levels of both PK and MK-4 have been shown to decrease γ-carboxylation of VKDPs [22].

Based on estimated dietary consumption, PK accounts for 50%, MK-4 makes up 10%, and MK-7, -8, and -9 represent 40% of total absorbed VK [23]. Being a fat-soluble vitamin, VK is taken up in the small intestine in the presence of dietary fat. A key mediator of intestinal VK absorption is Niemann–Pick C1-like 1 (NPC1L1) protein, a cholesterol and phytosterol transporter found in enterocytes and hepatocytes [24,25]. After absorption, PK is delivered to the liver and other tissues. It can be used unchanged, or it may be metabolized by certain types of microbiota into VK2 or into menadione in the human intestinal cells. A portion of menadione is transformed to MK-4, the dominant MK form in animal tissues [26]. However, there are tissue-specific VK distribution patterns. PK was found in all tissues with relatively high levels in the liver and heart but lower levels in the brain, lung and kidney. Compared to PK, MKs seem to be more important for extrahepatic tissues [27]. MK-4 levels were high in the brain and kidney and low in the liver, heart and lungs. The increased quantities of MK-4 in the brain suggest that this K vitamin is the active form of VK in this region [28]. Growing evidence advocates that MK-4 has a number of biological functions, including promoting growth factor of neuron-like cells, mediating apoptosis in several cancer cells, controlling glucose homeostasis [29]. In the central nervous system (CNS), MK-4 controls the activity of proteins involved in tissue renewal and cell growth control, myelination, mitogenesis, chemotaxis, neuroprotection [30]. The medium and long side-chain MKs were recovered mostly in the liver samples [31]. MK-7 and MK-4 converted from MK-7 increase collagen production and bone mineral density, promoting bone quality and strength [17]. As VK1, MK-4, and MK-7 have distinct bioavailability and biological activities, their recommended levels should be established based on their relative activities [32].

As present dietary intake recommendations are based on the dose required to prevent bleeding, novel data suggest that higher recommendations for VK consumption should be formulated [33]. Since both the bioavailability of VK from food as well as the endogenously produced VK are low, supplementation of VK should be considered for a number of chronic conditions, especially among elderly people [34].

Several scientific papers attested to the beneficial effects of VK in various chronic diseases, but supplement recommendations are difficult to outline. Nevertheless, a number of preclinical and clinical studies confirmed the safety of VK consumption. Several times higher dose levels than the estimated dietary intake for MK-7 did not show any toxicity in experimental animals [35]. In clinical studies, very high doses of MK-4 were used in the treatment of osteoporosis with no side effects [36].

The aim of this review was to summarize the recent scientific evidence on VK and its effect in preventing age-related diseases and/or improving the efficacy of some medical treatments in mature adults over 50 years old. To the best of our knowledge, it is the first study to concentrate on the effects of VK in this age group and to emphasize the role VK can play in the prevention of COVID-19.

## 2. Vitamin K in Bone Health

The musculoskeletal system, comprised primarily of muscle and bone, and the adipose tissue are connected through biological mechanisms underlying the physiological and pathophysiological crosstalk among muscle, bone, and fat [17]. Thus, several myokines (interleukin-6 (IL-6), myostatin) secreted by muscle have been identified as having effects on bone. Osteokines, especially OC, has been shown to have an endocrine impact on muscle, while adipokines (leptin, adiponectin, resistin) could act on either muscle or bone [37]. An in vitro study revealed that both carboxylated OC (cOC) and undercarboxylated OC (ucOC) increased secretion of adiponectin and the anti-inflammatory cytokine IL-10 and also inhibited secretion of tumor necrosis factor-α (TNF-α), but only cOC suppressed inflammatory IL-6 cytokine [38].

Thus, modifiable risk factors, such as healthy diets and physical activity, can positively affect these tissues. The role of calcium and vitamin D (vitD) in preventing osteoporosis is well established. However, more recent evidence suggests that other foods, such as fruit and vegetable, may have an essential role in bone health. Physical activity contributes to bone health by increasing serum total OC (tOC) and adiponectin, reducing leptin, and lowering insulin resistance [39].

Bone strength is determined by bone mineral content (BMC) and its quality and is associated with biological senescence and vitamin (B, D, K) deficiencies. As VK activates tissue-specific VKDPs, such as prothrombin, OC, or MGP, via the γ-carboxylation of Glu to Gla molecules, insufficient VKDPs γ-carboxylation is a sensitive, tissue-specific marker of VK deficiency [40]. Several studies revealed that VK is involved in bone metabolism and inhibits bone resorption in a dose-dependent manner. Binkley et al. showed that more than 250 µg/d VK intake is required for γ-carboxylation of OC [41].

Circulation OC is a marker of bone turnover. Of the total amount of OC that is released into the circulation, 40 to 60% is ucOC. This fraction of OC, being sensitive to VK intake, is a marker for VK status, usually revealing a lower VK availability [42]. Low dietary VK consumption and a high proportion of ucOC are independent risk factors for bone fractures in mature populations [43,44,45,46,47].

Table 2. summarizes the studies that showed an association between VK intake and bone parameters in mature subjects.

In a study including 221 healthy women, VK intake was significantly and negatively correlated with ucOC [55]. Correspondingly, higher VK consumption was associated with beneficial effects on fracture risk and bone health. Following an increased dietary green leafy vegetable intake by consuming approximately 200 g/d, 30 healthy individuals substantially reduced serum tOC, ucOC, and ucOC:tOC levels, suggesting increased entry of OC into the bone matrix, improvement of bone quality and lower fracture risk [62].

Moore et al. investigated the association between circulating VK1 with fracture risk in a study, including osteoporosis, in postmenopausal women. The results showed that serum VK1 concentrations were significantly higher in the group with fewer fractures and negatively associated with fracture risk [61]. The results of a 3-year study had the same conclusions: subjects with low plasma VK1 concentration had significantly higher susceptibility for vertebral fracture, independently of BMD, compared to the high VK1 group [54].

Postmenopausal women with osteopenia who received 5 mg of VK1 supplementation daily for 4 years had a significantly lower rate of fractures (*p* = 0.04) [52].

Besides leafy vegetables, dried plums (*Prunus domestica* L.), a rich source of VK1, demonstrated bone-protective effects. In a study of 84 osteopenic, postmenopausal women, 65–79 years of age, daily consumption of 50 g of dried plums for 6 months revealed less total body, hip, and lumbar bone mineral density (BMD) loss compared with that of the control group (*p* < 0.05), which can be explained by the ability of dried plums to suppress bone turnover and inhibit bone resorption [63]. Dried plums are rich in VK, potassium and minerals that are important to bone metabolism [64]. Booth et al. assessed the spine and hip BMD change in healthy elderly subjects, and after three years of follow-up, the daily PK supplementation did not present any additional benefit to BMD. However, the level of ucOC, associated with increased risk of bone fracture in older adults, significantly decreased [51]. Similar to the previous study, Emaus et al. observed that the daily intake of 360 µg MK-7 for one year increased cOC and decreased ucOC serum levels (*p* < 0.001) [65]. Feskanich et al. showed that women aged 38–74 years with higher daily VK intake had lower serum concentrations of ucOC and a 30% reduction in the risk of hip fracture compared to women with an intake of less than 109 μg VK per day [66]. Equally, the prevalence of VK deficiency was found to be higher in older patients (mean age 80.0) with hip fractures than those without [60].

In an intervention study, the use of 150 μg VK1 per day, in combination with physiological relevant doses of genistein, an important isoflavone [67], vitD, and polyunsaturated fatty acids (eicosapentaenoic and docosahexaenoic acids), could reduce fracture risk, at least at the hip, and prevent osteoporosis in postmenopausal women [68]. On one hand, VK2 supplementation might enhance the efficacy of vitD in bone and muscle health, improve bone quality, and reduce fracture risk in osteoporotic patients. On the other hand, vitD enhanced the carboxylation of OC, thus promoting the incorporation of calcium into the bone matrix and supporting bone metabolism [69]. Increased vitD intake should be accompanied by VK and magnesium supplementation to prevent long-term health risks, including hypercalcemia, a calcium buildup leading to calcification of the blood vessels and eventually osteoporosis. Hypercalcemia is not a vitD hypervitaminosis but rather a VK deficiency and higher serum concentrations of ucOC that inhibit calcium absorption in the bones [70].

In clinical studies, combined administration of VK and vitD, plus calcium, improved BMD, bone quality and decrease fracture risk, demonstrating a positive synergistic effect on bone health [57,71]. In a group of 181 healthy postmenopausal women, between 50 and 60 years, after 3 years of daily treatment with VK1, in addition to vitD, calcium, magnesium, and zinc, the bone loss at the site of the femoral neck was significantly reduced compared to the placebo group [72]. Furthermore, the results of clinical studies involving osteoporotic women of different ethnicity suggested that MK-4 in combination with calcium may be a safe approach in the treatment of osteoporosis [48,49,56,73].

Cockayne et al. investigated the effect of VK1 (1–10 mg/day) or MK-4 (15–45 mg/day) supplementations and showed that daily supplementation of MK-4 (45mg/day) reduced vertebral fractures (odds ratio (OR) = 0.40; 95% CI: 0.25–0.65), nonvertebral fractures (OR = 0.19; 95% CI: 0.11–0.35), and hip fractures (OR = 0.23; 95% CI: 0.12–0.47) compared to placebo and that MK-4 is a more effective antiosteoporotic agent than VK1 [74]. Hirao et al. observed that among postmenopausal women, who received osteoporosis monotherapy (alendronate, 5 mg/day) combined with 45 mg/day MK-4, over a period of one year, had a significant decrease in ucOC and ucOC:tOC ratio and reduced fracture rate compared with women, who received only alendronate monotherapy, suggesting that osteoporosis therapy could be improved with MK-4 supplements [53].

In another intervention study among healthy, non-osteoporotic women, the intervention group received 45 mg/day of MK-4 for three years and was compared to placebo. BMD did not change in the treatment group, though serum concentrations of tOC, cOC, and BMC significantly increased, maintaining bone strength at the site of the femoral neck. However, bone strength decreased significantly in the placebo group. Even at the very high doses of MK-4 used the adverse side effects [50].

The MK-7 isoform revealed the same benefits on bone health. In a 3-year randomized study, including healthy postmenopausal women, a daily supplement of MK-7 lowered circulating ucOC by ~50% and led to a significant improvement in bone density and bone strength [58].

Recent evidence showed that VK2 controls osteoblastogenesis and osteoclastogenesis via the NF-кB signal transduction pathway [75]. NF-кB signaling could, on one hand, inhibit osteoblastic differentiation and activity and, on the other hand, stimulate osteoclastic bone resorption. VK2 presented pro-osteoblastic and anti-osteoclastogenic actions, accomplished by downregulating inflammatory cytokines (e.g., TNF-α, IL-1) and inhibiting the activation of NF-кB [76]. This new mechanism explains the dual pro-anabolic and anti-catabolic activities of VK2 on bone. However, as no anti-NF-кB activity was associated with VK1 in this study, other mechanisms of action may be involved in the VK1 activity [75].

Liang et al. showed that BMD was significantly negatively associated with homeostatic model assessment for insulin resistance (HOMA-IR) and positively related with fasting glucose in the elderly population, suggesting that bone mass could be a predictor of glucose metabolism [77].

Several biological mechanisms may be involved in the prevention and treatment of aging-associated musculoskeletal disorders, including sarcopenia, osteoporosis, and osteoarthritis (OA). Clinical studies and animal experiments suggested an association between plasma VK status with muscle mass and strength, the link between the GGCX activity and bone protection, or the association between the steroid and xenobiotic receptor (SXR), a putative receptor for vitamin K, and cartilage protective effect [78]. Since VKDPs, including MGP, Gla-rich protein (GRP), periostin, and OC, were detected in cartilage and bone, VK may have a protective role in OA and joint health [79]. Thus, sufficient dietary VK intake and/or supplementation seemed to protect the population from age-related musculoskeletal diseases [80].

## 3. Vitamin K in the Prevention and Therapy of Vascular Calcification and Cardiovascular Diseases

Aging and several pathologic states, such as obesity, diabetes, or chronic kidney disease (CKD), cause degenerative changes of the vascular walls, including inflammation and vascular calcification (VC), leading to arterial stiffening and increased cardiovascular (CV) morbidity and mortality [81].

Ample evidence has shown that VK deficiency is related to the pathogenesis of VC [81,82,83,84]. VK has been suggested to inhibit VC and protect against cardiovascular disease (CVD) through the activation of VKDPs, such as MGP. To accomplish its potent calcification inhibitory function, MGP, secreted in the inactive form, needs activation (carboxylation), which takes place in the presence of VK. Upon activation, MGP binds calcium with high affinity, thereby inhibiting the VC process [82].

VC, a hallmark of senescence and a strong predictor of CV events, is another chronic inflammatory state induced via the generation of proinflammatory cytokines and mediated by the NF-кB signaling pathway. A high VK status may exert anti-inflammatory effects and prevent VC through antagonizing NF-кB signaling [83]. Growing evidence shows that VK as well as nuclear factor erythroid 2–related factor 2 (Nrf2) signaling could play a vital role in blocking ROS generation, cellular senescence, DNA damage, and inflammaging [84].

In CKD, a pathological condition characterized by osteoporosis, sarcopenia, and increase CVD events [85], VC is widespread even at early stages. Besides careful attention to calcium and phosphate balance, no particular therapy enabling regression or inhibiting the progression of VC existed [86]. Accumulating evidence describes the VC mechanism as an active process involving calcification promoters and inhibitors. The biologically active MGP, highly dependent on VK status, is viewed as a strong inhibitor of vascular elastic fiber damage and VC [87] and also the only factor that can actually reverse the process [88]. The inactive, uncarboxylated form of this protein reflected the deficiency of VK status and has been linked with VC and CV events. Growing scientific data show that VK-dependent MGP could offset age-related wear and tear on the arteries, VC, and CVD development [89].

To date, a number of experiments and observational studies examined the effects of VK supplementation and dietary intake on vascular calcification and CVD (Table 3) in mature populations.

Several studies demonstrated that higher dietary consumption of VK2 significantly reduced the incidence of VC and coronary heart disease (CHD) [90,91]. In these studies, no association between VK1 intake and CHD was detected while controlling for confounders. After monitoring 2987 participants during a median follow-up time of 11 years, only dietary MKs, but not VK1 intake, were significantly associated with a lower risk of CHD [117]. Scientific evidence specified that VK1 mainly carboxylate VK-dependent factors in the liver, while VK2 is responsible for the carboxylation of VKDPs in the extrahepatic tissues [118]. Nonetheless, it was demonstrated that higher doses of VK1, namely 2 mg/d, can also act in extrahepatic tissues and delay the progression of VC [110]. Furthermore, low plasma VK1 status was linked with higher all-cause mortality risk [115] and with an increased risk for CVD in older patients treated for hypertension [111].

VK intake slowed the progression of preexisting coronary artery calcification (CAC), a well-known independent predictor of CVD risk, in asymptomatic older men and women [92]. Moreover, adequate consumption of VK-rich foods has been suggested as both preventing action and prospective adjuvant therapy against atherosclerosis and stroke [116].

A combination of low VK and vitD status is associated with the increased left ventricular mass index, a parameter for cardiac structure, which has been shown to predict higher mortality, as well as the augmented risk of all-cause mortality in older populations [113]. In diabetic patients with stable CHD, combined supplementation with MK-7, vitD, and Ca was associated with a significant reduction in maximum levels of left carotid intima-media thickness (a parameter positively linked with diabetes, blood pressures, lipid profiles, inflammatory cytokines), C-reactive protein (CRP) and malondialdehyde (MDA) levels, and a significant increase in high-density lipoprotein (HDL)-cholesterol levels [105].

A functional VK deficiency is strongly associated with an increase in uncarboxylated VK-dependent protein levels, the hepatic protein induced by vitamin K absence-II (PIVKA-II) and extrahepatic dephosphorylated-uncarboxylated matrix Gla protein (dp-ucMGP) [99]. Scientific findings reported that VK could modulate dp-ucMGP levels and that plasma dp-ucMGP levels decline after VK intake in a dose-dependent manner [97,100]. Circulating plasma dp-ucMGP levels augmented progressively in many diseases and were directly correlated with the severity of VC, cardiac function and long-term mortality [93,94,95,96]. Equally, in a study involving 2318 subjects, elevated dp-ucMGP increased the risk of CV (*p* = 0.027) and all-cause (*p* ≤ 0.008) mortality [119]. Similarly, in diabetes patients with high CV risk, elevated levels of dp-ucMGP and lower levels of total ucMGP (*t*-ucMGP) are independently related to the severity of peripheral artery calcification [101]. Moreover, higher dp-ucMGP values were independently associated with carotid-femoral pulse wave velocity (cfPWV) in diabetes and CKD patients and may lead to large arterial stiffening [108,112].

Adequate dietary intake of VK may be essential in reducing atherosclerosis progression, CV risk, or CVD and all-cause mortality in CKD patients [102,104,107]. CKD and hemodialysis patients could often present vascular VK deficiency due to significantly low VK intake, resulting in an elevated risk of VC and bone fractures [120]. After three years of 180 µg MK-7 daily intake, dp-ucMGP levels decreased by 50% compared to placebo [103]. Even after a shorter 12 week-period, ucMGP, an independent risk factor for arteriosclerosis and CVD, significantly decreased in the MK-7 supplementation groups compared to placebo [98]. Other interventions with different amounts of MK-7 (100 µg/d and 360 μg/d) provided significant effects on dp-ucMGP [106,109].

Diabetic CKD patients with plasma dp-ucMGP levels above the median (≥ 656 pM) had a significantly higher risk for CV events, CV mortality, and all-cause mortality compared to the low dp-ucMGP group [114]. High levels of dp-ucMGP were significantly associated with higher triglycerides (*p* = 0.03) and C-reactive protein (*p* = 0.03) levels, CV mortality (*p* = 0.037), all-cause mortality (*p* = 0.02), and progression of CKD (*p* = 0.024) [114]. Likewise, a prospective study investigating 4275 people (aged 53 ± 12 years, 46.0% male) for 10 years, concluded that plasma dp-ucMGP was associated with total (hazard ratio (HR) = 1.14; 95% CI: 1.10–1.17, p ≤ 0.001) and CV (HR = 1.17; 95% CI: 1.11–1.23, p ≤ 0.001) mortality [121].

Recent data indicated that dp-ucMGP levels might be associated with high-risk for CV mortality and all-cause mortality. One meta-analysis, which included 11 studies and 33,289 patients, revealed that high circulating dp-ucMGP was associated with increased risk of all-cause and CV mortality [122]. Correspondingly, another large meta-analysis comprising 21 articles and 222,592 subjects exposed that elevated plasma dp-ucMGP levels were correlated with higher risk of all-cause mortality (HR = 1.84; 95% CI: 1.48–2.28, *p* < 0.001), CVD mortality (HR = 1.96; 95% CI: 1.47–2.61, *p* < 0.001), as well as increased total CVD risk (HR = 1.57; 95% CI: 1.19–2.06, p < 0.001) [123]. This study also found a significant association between dietary VK1 and MKs with total CHD (HR = 0.92 and 0.70, respectively), but no correlation was noticed between dietary VK and all-cause or CVD mortality [123].

In conclusion, as no toxicity or serious side-effects of VK intake have been described, even for higher doses, patients with CVD risk could benefit from VK supplementation, a safe therapy, which can present significant clinical impact.

## 4. The Effects of Vitamin K on Metabolic Disorders

Obesity and type 2 diabetes (T2D) are metabolic disorders affecting the world population with serious health and economic complications. Obesity, as well as overweight, is a risk factor for deficiency of fat-soluble vitamins. Data reported that VK supplementation reduced OS, insulin resistance, and lowered progression of metabolic risk biomarkers for T2D. There was a clear association between circulating VK and dependent-OC concentration, obesity and T2D risk [124]. Scientific evidence suggests that OC, an osteoblast-derived hormone, is involved in glucose and energy metabolism through multiple mechanisms. It regulates secretion and insulin sensitivity through increase β-cell function and increases adiponectin expression in adipocytes. Metabolic disorders, including obesity or diabetes, can affect the synthesis and action of OC, causing a disruption of the bone–energy metabolism axis [125].

Dietary patterns stressing plant food consumption may be effective in both preventing T2D and improving diabetes management. VK may play an imperative role in the regulation of glycemic status by improvement of insulin sensitivity, which may decrease the risk for T2D [126].

The synergistic effect exerted by the bioactive molecules (e.g., lipophilic vitamins, such as VK) found in plant or animal source foods can improve insulin sensitivity through a number of signaling pathways in the prediabetic and diabetic population [127]. VK may regulate glucose levels through controlling OC levels and inflammation and exert beneficial effects in T2D [128].

The design and outcomes of studies assessing the effects of VK supplementation on metabolic disorders are shown in Table 4.

The relation between OC and energetic metabolism was assessed in a cross-sectional study, including 146 postmenopausal women with and without T2D. Diabetic women presented lower levels of serum tOC (*p* < 0.05). There were significant negative correlations between OC concentration and glycated hemoglobin (HbA1c), serum triglycerides, and body mass index (*p* for all < 0.05), independent of the presence of T2D [139].

Similarly, in a study carried out in postmenopausal non-osteoporotic women, OC was found to be significantly lower in women with metabolic syndrome (metS) compared to control (*p* < 0.001). In this study, a significant positive correlation (*p* = 0.008) was detected between vitD and OC [140]. Supplementation of vitD, vitD3 metabolite more than vitD2, revealed to have favorable effects on metabolic profile measurements and depressive symptoms in T2D patients [144].

Apparently, VK supplementation had no significant consequences on glycemic control in healthy subjects [145]. However, studies performed on prediabetic and diabetic patients to determine the VK effect had different results. Rasekhi et al. studied 82 premenopausal and prediabetic women (40.17 ± 4.9 years), who were randomized to consume either 1000 µg PK supplement or placebo in a randomized controlled trial. After 4 weeks, the PK intake increased the serum levels of cOC and decreased ucOC compared with placebo (for both: *p* < 0.001) and improved the insulin sensitivity. A statistical significant association between changes of ucOC and 2 h post-oral glucose tolerance test (OGTT) glucose was found (*r* = 0.308, *p* = 0.028) [146].

Among patients with metS and T2D, both VK forms were beneficial. However, the risk reduction occurred at higher levels of PK intake compared to MK, suggesting that MK could be more effective than PK in reducing T2D risk [147]. It seemed that MK improved insulin sensitivity through the contribution of OC, anti-inflammatory activity, and lipid-lowering effect [148].

In a 12 week-trial involving T2D patients, the intake of 200 µg MK-7 daily supplements significantly decreased fasting blood sugar (*p* = 0.02) and HbA1c (*p* = 0.01) compared to the placebo group. Although MK-7 supplementation improved glycemic indices, the lipid profile did not significantly change within or between groups [149]. The same parameters were investigated in another randomized controlled trial (RCT), with a higher intake of MK-7, 180 µg twice daily. After 12-weeks, the T2D patients in the MK-7 group had significantly lower levels of fasting plasma glucose and HbA1c compared with the placebo group, while again, no significant changes were noticed in the lipid profiles. Fasting insulin and HOMA-IR significantly decreased in the MK-7 group compared to baseline, suggesting a decrease in insulin resistance [143].

The MK-7 isoform intake was yet again analyzed in another trial for eight weeks. A number of 84 polycystic ovary syndrome (PCOS) patients were randomly assigned into the 90 µg MK-7 daily treatment group and placebo. At the end of the study, MK-7 supplementation significantly decreased serum fasting insulin (*p* = 0.002) and HOMA-IR (*p* = 0.002) compared to the placebo group. Furthermore, MK-7 intake led to significantly lower serum triglyceride level (*p* = 0.003), waist circumference (*p* = 0.03), and body fat mass (*p* < 0.001). In this study, MK-7 intake showed beneficial effects on glycemic indices but also on lipid and anthropometric profiles in PCOS patients [150].

Knapen et al. assessed fat mass and body composition in postmenopausal women. The group that received 180 µg MK-7 per day revealed higher levels of circulating cOC. In subjects with an above-median response in cOC, a significant increase in adiponectin level and a decrease in abdominal fat mass and visceral adipose tissue area were observed compared with the placebo group and the subjects with low cOC level. Thus, MK-7 intake could reduce body weight or abdominal and visceral fat in subjects showing a strong increase in cOC [138].

A study, which evaluated the effect of vitD3 and VK2 supplements alone or in combination on OC levels and metabolic parameters was conducted in 40 diabetic patients. Diabetic patients are characterized by bone demineralization and changes in OC levels. In the vitD3 plus VK2 group, a significant decrease in glycemia (*p* = 0.002), percentage of pancreatic β-cells (*p* = 0.004), and in the uOC/cOC index (*p* = 0.023) were noticed. In the VK2 group, again a significant decrease in glycemia (*p* = 0.002), percentage of pancreatic β-cells (*p* = 0.002), and HOMA‑IR (*p* = 0.041), and a statistically significant increase of cOC concentrations were observed. The increase in the cOC concentration could be explained by the action VK2 as a cofactor of carboxylases during activation of OC [141].

Yoshida et al. analyzed PK supplementation in an RCT comprising 355 nondiabetic men and women (mean age 68 years). After 36 months, HOMA-IR was significantly lower among men in the 500 μg PK daily supplement group compared to the control group, but no statistically significant result differences were noticed in women. Thus, PK supplementation for three years decreased the levels of ucOC and had a protective effect against the insulin resistance progression in older men [151]. In older humans, serum cOC and not ucOC concentration was associated with lower insulin resistance [133], which supports a potential link between bone physiology and insulin resistance in humans.

Jeannin et al. explored the association between VK status and diabetic peripheral neuropathy. The levels of dp-ucMGP, an inverse marker for VK status, were significantly higher in patients with neuropathy versus patients without neuropathy (*p* = 0.009). Since dp-ucMGP is a VK-dependent protein, reduced VK status is an independent risk factor for diabetic peripheral neuropathy. Hence, treatment with VK supplements may be a preventive measure in diabetic patients at risk of peripheral neuropathy [142].

VK consumption was linked with increased cOC, in addition to improved glycemic status, dyslipidemia, serum insulin, OS, and inflammation in T2D [152]. Possible mechanisms of these effects could be reduced hepatocyte gluconeogenesis and lipogenesis, decreased production of inflammatory cytokines and higher levels of adiponectin, inactivation of NF-кB pathway, or increased gene expression levels of AMP-activated protein kinase (AMPK) and sirtuin-1 (SIRT-1), important signaling molecules in the regulation of glucose hemostasis, lipid metabolism, and insulin sensitivity [152,153].

In animal studies, ucOC was found to be the active hormonal form that conferred beneficial glucose control and the only molecule involved in the production of insulin by the pancreatic β-cells [154]. Opposite to what was proposed in mouse models, in humans, the association between ucOC and insulin resistance may differ [155]. Higher VK intakes and increase cOC were associated with a low percentage of ucOC but also with reduced blood glucose, insulin resistance, and T2D risk [130,156,157]. The outcomes in these human studies assumed that a low percentage of ucOC actually improves glucose metabolism. Moreover, both cOC and ucOC levels could increase glucose transport in adipocytes and muscle cells and improve insulin sensitivity [38]. Although the in vivo experiments could have remarkable value in human pathology studies, some animal models cannot be extrapolated directly to humans [158].

Based on the current literature, healthier dietary habits and lifestyle, such as consumption of green leafy vegetables and fermented foods, major sources of VK, may independently contribute to reducing metabolic disorder risks.

## 5. The Effect of Vitamin K on Neurodegenerative Diseases

Age-related neurodegenerative diseases, such as Alzheimer’s disease (AD) or Parkinson’s disease (PD), lead to one of the most unfavorable health problems, cognitive impairment. It is a legitimate age-related health concern potentially affecting the wellbeing and independence of mature and old adults [159]. The dysregulations in these pathologies are mainly associated with OS, neuroinflammation, abnormal protein aggregation, or mitochondrial dysfunction. Recent animal and human studies showed that bioactive compounds could diminish the risk or delay the onset or progression of inflammation processes, cognitive impairment, or age-related syndromes [160,161,162].

Healthy nutritional diets, modifiable lifestyle factors may prevent or delay these diseases. Increased consumption of vegetables, fruits, nuts, seeds, with proven antioxidant and anti-inflammatory activities, is the principal dietary recommendation, with an important reminder that the beneficial effects may come from wholesome, healthy diets rather than from a particular nutrient [163].

AD, described by the existence of intracellular neurofibrillary tangles containing the microtubule-associated protein tau and extracellular aggregated amyloid-β (Aβ) peptides, is the most common cause of dementia in the old population. These modifications induce a chronic inflammatory state, leading to the neuronal damage observed in AD [164].

New findings suggest the participation of VK in brain physiology through the carboxylation of Gas6, a VKDP, which could defend against neuronal apoptosis induced by OS and Aβ [165]. Moreover, VK is implicated in neuron development and survival, which are mediated by protein S and sphingolipids. Sphingolipids are a class of lipids extensively present in brain cell membranes with important cell roles. They are active in neuroprotection and myelination, a critically important process for healthy CNS functioning [166]. VK may reduce cognitive decline and the risk of AD through modulating sphingolipid metabolism, which leads to enhanced Aβ clearance [166]. Altered sphingolipid profiles have been linked to neuroinflammation and neurodegeneration [167].

Recent evidence has shown that during remyelination, VK enhances the production of brain galactosyl ceramides, cerebrosides with a major role in nerve cell membranes. Furthermore, VK appears to have a survival-supporting effect on neurons [142].

Fat-soluble vitamins (A, D, E, and K) or water-soluble vitamin C are powerful antioxidant and anti-inflammatory agents [168]. Inadequate concentrations of vitamins have been linked with brain aging and cognitive decline in AD patients and the elderly [169]. VK has been shown to influence AD risk and cognitive functions, positively impact the mechanisms involved in AD pathogenesis, including OS, inflammation, Aβ-aggregation and Aβ-induced neurotoxicity [170].

Low plasma VK concentration was correlated with a greater degree of frailty, common in patients with neurodegenerative diseases. The relationship between VK status and frailty was assessed in a longitudinal study with 644 (54% women) community-dwelling adults, mean age 59.9 years over 13 years. After measuring dp-ucMGP as a marker of VK status, compared with the lowest tertile, the medium (1.40; 95% CI: 0.01–2.81, *p* for trend = 0.03) and highest (1.62; 95% CI: 0.18–3.06, *p* for trend = 0.03) tertiles were associated with higher degree of frailty [171].

Data reported low serum VK concentrations in AD patients and disclosed that patients with early-stage AD consumed lower VK per day than cognitively intact control subjects, which consumed around 139 μg VK daily [172]. Likewise, results from a mature population, 65 years and older, revealed a direct correlation between low VK dietary intake and low serum VK concentration, as well as declined cognitive performances [173]. Some MK isoforms, mainly the longer chains, produced by the gut microbiota were positively associated with cognition, as demonstrated by McCann et al. in a study on 74 old individuals at different cognitive ability levels [174].

The concentration of circulating PK is positively correlated with the intake, as it was demonstrated in a representative sample population aged over 65 years [175]. Tanprasertsuk et al. showed that in a group of nondemented centenarians, only circulating PK levels were significantly linked with a wide range of cognitive performance. Despite the fact that MK-4 was the predominant isomer in both the frontal and temporal cortex, cerebral MK-4 levels were not associated with cognitive measures. VK-rich food intake containing other bioactive molecules may act in synergy to cognitive health [176]. In a cross-sectional study, which comprised 320 old participants aged 70 to 85 years and without cognitive impairment, higher serum levels of PK were significantly connected with better verbal episodic memory performances (*p* = 0.048), exposing better cognition during aging [177].

The results of a prospective study that included 960 subjects (mean age 80 years) revealed that the intake of at least one serving of PK-rich foods daily, including green leafy vegetables, was linked with slower cognitive deterioration, corresponding to 11 years younger in age for the subjects in the highest quintile of PK intake (median 1.3 servings/d) [178]. Similarly, Chouet et al. indicated a statistically significant association between increased dietary PK intake and better cognition and behavior [179]. In a group of 192 participants (mean age 83 years), cognition was assessed with the mini-mental state examination (MMSE) and behavior with the frontotemporal behavioral rating scale (FBRS). Compared to lower intake, participants with higher PK intake had greater (i.e., better) mean MMSE score (22.0 ± 5.7 vs. 19.9 ± 6.2, *p* = 0.024) and lower (i.e., better) FBRS score (1.5 ± 1.2 vs. 1.9 ± 1.3, *p* = 0.042) [179].

Evaluating MS patients, Lasemi et al. showed that MK levels in this population were decreased compared to controls and suggested that MK supplementation might inhibit the disease’s evolution [180]. Indeed, Sanchez et al. observed that prophylactic MK supplementation could suppress experimental autoimmune encephalomyelitis, an animal model of brain inflammation used to study human CNS demyelinating diseases, including MS [181].

A relationship between PD and serum VK2 levels was examined by Yu et al. in a study involving 93 PD patients and 95 healthy controls (age over 66) [182]. The results indicated that the serum VK2 level of PD patients was significantly lower (3.49 ± 1.68 ng/mL) than that of healthy controls (5.77 ± 2.71 ng/mL). Since inflammation is important pathogenesis of PD, and VK has anti-inflammatory action, deficiency of VK may lead to occurrence and aggravation of inflammatory state, and eventually the incidence of PD [182].

Significantly lower dietary VK consumption was associated as well with serious subjective memory complaints in 160 studied patients (mean age 82 y). Patients with serious subjective memory complaint had lower mean dietary VK intake compared to participants without serious subjective memory complaint (298.0 ± 191.8 μg/day vs. 393.8 ± 215.2 μg/day, *p* = 0.005). Increased VK intake was linked with fewer and less severe subjective memory disorders in participants taking no VK antagonists (VKAs) [183]. The use of VKAs as anticoagulant medications lowered the VK bioavailability, thus reducing the VK concentration and increasing the altered cognitive performance risk and the frequency of cognitive impairment in the elderly [184,185].

Scientific evidence confirmed that OC is involved in multiple biological processes, including energy metabolism, cognition, stress response, CV health. These physiological functions have been documented to be regulated by both OC forms, cOC and ucOC [186]. OC can bind to neurons of the hippocampus, brainstem, or midbrain, and enhance the production of monoamine neurotransmitters, prevent anxiety and depression, and support learning and memory. During aging, a decline in bone mass may cause a decrease in cognitive functions because of a drop in OC synthesis and/or activation [187]. As bone function and cognitive features deteriorate in parallel with OC levels during aging, this molecule could be defined as an antiaging tool with the potential to be used against age-related disorders, including cognitive alterations. Improving bone health during aging may have favorable effects on cognition [188].

Since pharmacological interventions have been unsuccessful in the prevention of dementia and evolution of AD or PD, other approaches, such as lifestyle changes and dietary therapies, may impact the prevention and evolution of dementia, AD, or PD. Thus, nutritional interventions could favorably modulate the epigenetic mechanisms through regulating DNA acetylation and methylation or altering the expression of miRNAs [189]. Foods, including green leafy vegetables, berries, or nuts, high in VK and other vitamins, minerals, polyphenols with powerful antioxidant properties, should be encouraged in older adults for the prevention or management of age-associated neurodegenerative diseases [190].

Moreover, VK, especially VK2, prevents an excess vascular calcification of retinal blood vessels and thus the age-related stiffness and atherosclerotic plaque of blood vessels. It can stop the evolution of an age-related macular degeneration (AMD) and, for better results, should be used in combination with magnesium, zinc, and/or vitamin D [191].

## 6. The Effect of Vitamin K on Cancer

The basic chemical structure of VK and the functional unit in several cancer chemotherapeutic drugs is a quinone, which partially explains the research involving VK use in the prevention and treatment of cancer [192].

Quinones can be converted into reduced forms, first into intermediate semiquinones (one-electron reduction), then hydroquinones (two-electron reduction). These reactions consume superoxide radicals, generally accepted as oncogenic, and also consume reducing equivalents (NADH, NADPH, glutathione), essential for cancer cell homeostasis [193], hence creating an intracellular setting proper for induction of apoptosis. The VK-modulated redox-cycle may partially explain VK anticancer activity [194].

Further research suggests that increased VK intake (e.g., MK) may have potent anticancer properties since it has shown an inverse association with overall cancer incidence. Although the exact anticancer activity of dietary VK is still unclear, there are several suggested mechanisms that may explain its effect on preventing carcinogenesis, such as scavenging oxygen free radicals, inhibiting polyamine metabolism, induction of apoptosis, production of reactive oxygen species (ROS), cell cycle arrest and activation of antimetastasis genes [195,196].

Both VK1 and VK2 have demonstrated antiproliferative, proapoptotic, autophagic activities, resulting in anticancer activity [195]. Moreover, VK3 and its analogs are potent inhibitors of cell proliferation on many cancer cell lines. They act as cellular redox mediators generating ROS and inducing apoptosis by mitochondrial pathway [197]. Combining VK3 with other molecules sharing structural similarity, such as plumbagin and juglone, naturally occurring naphthoquinones found in polyphenol-rich *Juglans regia* [198], or with vitamins or drugs that also function through modulation of intracellular redox states could potentiate the antitumor effects [199,200].

Cancer cell death induced by VK2 appears to vary among the type of cancers. In triple-negative breast cancer cell lines, VK2-induced non-apoptotic cell death along with autophagy [201]. In prostate cancer cell lines, VK2-induced cell death through ROS-mediated cell cycle arrest and mitochondrial-mediated apoptosis, as well as metastasis-inhibiting signaling molecules [202]. Moreover, in prostate cancer cells, VK2 showed anti-inflammatory activity as several inflammatory genes were downregulated after treatment with VK2. Additionally, VK2-reduced proliferation, induced apoptosis and lowered the angiogenic potential of prostate cancer cells. The proposed mechanisms for the potential anticancer effects were caspase-3 induction, inhibition of NF-кB pathway, downregulation of phosphorylated protein kinase B (AKT), and reduction of androgen receptor expression [203]. Moreover, certain essential proteins, such as Bak and Cx43, several protein kinases, such as PKA and PKC, and transcription factors, such as AP-2, are involved in the mechanism of VK2 activity against cancer cells [204].

In different cancer cell lines, VK2 can inhibit cancer cells’ growth by the initiation of autophagy, a natural mechanism that removes damaged or dysfunctional cellular organelles and prevents diseases, such as cancer, diabetes, or neurodegeneration [205]. Yokoyama et al. demonstrated that MK-4 could simultaneously stimulate autophagy and apoptosis in leukemic cells, but rather autophagy was dominant in the presence of B-cell lymphoma 2 (Bcl-2) protein that inhibited apoptosis [206]. Similarly, MK-4-treatment-induced antitumor effects on cholangiocellular carcinoma (CCC) cells via autophagy. The apoptosis induction effect of MK-4 in CCC cells was relatively small compared to other cancer cells, a possible reason being, as in the previous experiment, the over-expression of the anti-apoptotic Bcl-2 protein in CCC cells [207]. Tokita et al. examined the growth inhibitory action by MK-4 on gastric cancer cell lines. The results established that MK-4-treatment-induced antitumor effects through apoptosis and cell cycle arrest in a dose-dependent manner [208].

In another study, MK-4 again inhibited the growth and invasion of hepatocellular carcinoma (HCC) cells via activation of protein kinase A. MK-4 reduced the ability of liver cancer cells to invade and spread via the portal venous system [209]. However, the beneficial effects of VK treatment alone were not enough to avoid or treat HCC in clinical settings. Thus, VK administration combined with other anticancer reagents could achieve satisfactory therapeutic effects against HCC [210]. Yoshiji et al. administered MK-4 (45 mg/d) and angiotensin-converting enzyme inhibitor (ACE-I) (4 mg/d) after curative therapy for HCC. After 48 months, the combination treatment with VK and ACE-I inhibited the cumulative recurrence of HCC, at least partly through suppression of the vascular endothelial growth factor (VEGF), an angiogenic factor [211].

Duan et al. examined the anticancer activity of VK2 in bladder cancer cells and investigated the underlying mechanism. VK2-induced apoptosis in bladder cancer cells through the phosphorylation of c-Jun N-terminal kinase and p38 mitogen-activated protein kinase (JNK/p38 MAPK), as well as through mitochondrial pathways, including loss of membrane potential, cytochrome C release, and caspase-3 cascade [212]. Furthermore, VK2 can upregulate glycolysis in bladder cancer cells, mediated by phosphatidylinositide-3-kinase and AKT (PI3K/AKT) and hypoxia-inducible factor-1α (HIF-1α), induce metabolic stress, along with increased phosphorylation of AMPK and reduced phosphorylation of mammalian target of rapamycin complex 1 (mTORC1). Thus, in response to metabolic stress, VK2 could activate AMPK and suppress the mTORC1 pathway, consequently causing AMPK-dependent autophagic cancer cell death. Upon glucose limitation, the increased glycolysis would result in metabolic stress and cell death. Hence, VK2 could induce metabolic stress and trigger AMPK-dependent autophagic cell death in bladder cancer cells by PI3K/AKT/HIF-1α-mediated glycolysis elevation, this being one of the VK2-induced anticancer mechanism [213].

Dietary polyamines are involved in various biological processes, including cell proliferation and differentiation, which can increase life span and be beneficial against aging and age-related disorders [214,215]. However, polyamines are detrimental in disease progression and are a target for anticancer agents [215]. PK proved to be a potential anticancer agent. Following PK administration to colon cancer cell lines, significant antiproliferative and proapoptotic effects were noticed, in addition to a significant decrease in the polyamine biosynthesis [216].

The association between dietary intake of PK and MKs and total and advanced prostate cancer was evaluated in 11,319 men during a mean follow-up time of 8.6 years. MKs intake presented a nonsignificant inverse association for total prostate cancer (RR = 0.65; 95% CI: 0.39–1.06) and a significant association for advanced prostate cancer (RR = 0.37; 95% CI: 0.16–0.88, *p* for trend = 0.03). The association was stronger for MKs from dairy products compared with MKs from meat. The PK intake did not correlate with prostate cancer incidence [192].

VK has been reported to have antiproliferative and proapoptotic activity in human melanoma cells. VK3 has been identified as a specific inhibitor of the E3 ubiquitin ligase Siah-2, an enzyme implicated through several mechanisms in melanoma development and progression [217].

Beaudin et al. reported distinct effects on breast cancer cells for the two forms of VK, as VK1 promoted γ-carboxylation and stem cell features, while VK2 presented antiproliferative or proapoptotic effects. The authors hypothesized that in normal breast, VK1 is converted to VK3, which is then prenylated by the enzyme UbiA prenyltransferase domain containing 1 (UBIAD1) to VK2, favoring tumor suppression. However, loss of UBIAD1 in tumors abrogates VK2 formation, leading to accumulation of VK1, which promotes aggressive phenotypes via γ-carboxylation if tumors express the enzyme GGCX. Future studies could clarify the function of UBIAD1 and the action of cellular VK1 and VK2 in breast cancer cells [218].

Contrary to the previous hypothetical opinion, the data from a large prospective cohort study showed that dietary VK2 was linked with breast cancer incidence and mortality. After adjustment for confounders, total VK and dietary VK1 were not associated with breast cancer incidence and mortality. However, total VK2 intake was significantly associated with 26% elevated breast cancer risk, and 71% increased risk of death from breast cancer [219]. In the general population, VK2 intake is mainly from cheese and meat and, based on recent scientific evidence, meat consumption and not VK2 was associated with increased breast cancer risk [220]. Other prospective studies found an association between better diet quality and higher consumption of salad vegetables, rich sources of VK1, and lower risk of breast cancer, offering indirect evidence for the antioncogenic effect of VK1 [221,222].

In the prospective European Prospective Investigation into Cancer and Nutrition—Heidelberg cohort study, 24,340 participants were followed for more than 10 years to estimate an association between VK intake and overall cancer incidence and mortality. Dietary intake of VK2, highly determined by cheese consumption, was significantly inversely associated with cancer mortality (HR = 0.72; 95% CI: 0.53–0.98, p for trend = 0.03) and nonsignificantly linked with overall cancer incidence (HR = 0.86; 95% CI: 0.73–1.01, p for trend = 0.08) for the highest compared with the lowest quartile. Cancer risk reduction after VK2 intake was more evident in men than in women, mostly driven by significant inverse associations with lung (p for trend = 0.002) and prostate (p for trend = 0.03) cancer. In women, almost 50% of all cancer cases were breast cancer, nonsignificantly associated with VK2 intake [223]. Dietary VK2 intake was more strongly inversely associated with cancer mortality than with cancer incidence because likely, factors having a role in apoptosis and cell cycle arrest appear later in carcinogenesis. In addition, the suggested VK2 inhibitory role in angiogenesis is strongly linked to metastasis development [224].

Another prospective cohort, the PREDIMED study, which enrolled 7216 participants with high CVD risk, means age 67 years, followed up for a median of 4.8 years, analyzed the link between dietary VK intake and cancer risk, among other parameters. After adjustment for potential confounders, the outcomes indicated that dietary VK1 intake was associated with a significantly reduced risk of cancer (HR = 0.54; 95% CI; 0.30–0.96). In longitudinal analyses, individuals who increased their intake of PK or MK during follow-up had a significantly lower risk of cancer (HR = 0.64; 95% CI; 0.43–0.95; and HR = 0.41; 95% CI; 0.26–0.64, respectively) compared to individuals, who diminished or did not change the VK intake. Thus, dietary intake of both PK and MK forms was associated with a reduced risk of cancer, besides a lower risk of CV and all-cause mortality [225]. Although in this study PK was positively correlated with cancer risk, in other studies, dietary PK intake was not associated with cancer. Considering that PK is converted to menadione and MK-4, it can be proposed that dietary PK exerts cancer inhibitory effects as part of the total VK concentration [223].

The universal agreement is that a healthy vegetable-rich diet could prevent cancer and its development. Thus, the protective effects of a high PK consumption on carcinogenesis may come from healthy diets with beneficial synergistic effects rather than from VK per se.

## 7. Correlation between Vitamin K and Pulmonary Disease

The most common chronic respiratory disease is a chronic obstructive pulmonary disease (COPD) involving chronic bronchitis and emphysema. In a cross-sectional study, the association of dark green vegetables with emphysema status was assessed among US adults. The consumption of recommended amounts of VK was associated with a 39% decrease in odds of emphysema. VK showed that it might slow the emphysematous process and, together with vitamin A are important in lung health [226].

VK can activate intrahepatic and extrahepatic procoagulant or anticoagulant factors, such as protein S. This protein, a VK-dependent plasma glycoprotein, has a role in the anti-coagulation pathway, where it functions as a cofactor to protein C [227]. Besides this action, protein S can prevent the production of inflammatory cytokines associated with the cytokine storm observed in acute lung injury [228]. Alterations in the serum levels of protein S can relate to the progression of fibrosis and inflammatory diseases in the lung, liver, or heart [229]. Low protein S levels were correlated lately with higher thrombogenicity, clinical severity, and fatal outcome in COVID-19 patients, independently of age or even Inflammatory biomarkers [230]. In COVID-19 cases, the reduced activation of MGP and protein S due to the pneumonia-induced VK depletion can lead to an escalation in pulmonary injury and thrombosis [231].

## 8. Conclusions

The latest scientific evidence summarized in this review indicated that VK has a significant role in mitigating aging and preventing age-related diseases and has the potential to improve the efficacy of some medical treatments among adults over the age of 50 years. The novel role of VK on aging and age-associated diseases is mainly due to its antioxidant and anti-inflammatory effects. The review focused on the most prevalent age-related diseases, including osteoporosis and bone fractures, neurodegenerative diseases, VC, CVD, and cancer, as well as metabolic disorders, mainly T2D and obesity. In addition, we presented the most recent findings on the association between VK and COVID-19 and its potential effect on reducing fatal outcomes in such cases. Specifically, the scientific data showed that VK has an integral role in bone metabolism through the carboxylation of OC, which is an important protein capable of transporting and depositing calcium in bone. MK-4 was revealed to be a more effective antiosteoporotic agent than VK1, with increased pro-osteoblastic and anti-osteoclastogenic actions achieved by inhibiting the NF-кB pathway. VitD improves OC carboxylation and, along with VK and magnesium supplementations, can be a better strategy for reducing bone fractures, a highly public health concern among the elderly. In addition, the review concluded that VK supplement could be a safe approach for reducing CVD morbidity and mortality. By activating matrix Gla protein, VK keeps calcium from accumulating in the walls of blood vessels, thus making VK a potential treatment for patients at risk for either VC or CVD. Furthermore, VK may reduce the risk for metabolic disorders, such as T2D, by improving insulin sensitivity and anti-inflammatory activity, as well as obesity, through a lipid-lowering effect. The review also showed the influence VK has on age-related neurodegenerative diseases, such as AD and PD. VK is involved in the brain’s physiology and can reduce its cognitive decline by carboxylation of Gas6 protein, a VKDP that could defend against neuronal apoptosis induced by OS and Aβ. The anticancer potential of VK was summarized by reviewing several in vitro and epidemiological studies. There are multiple mechanisms where the potential anticancer agent of VK can react, including the modulation of various transcription factors, which induced antiproliferative, proapoptotic, and autophagic effects, which were found to be associated with a reduced risk of cancer. The latest evidence on VK and pulmonary disease stem from the fact that VK can activate protein S, which was recently shown to prevent the generation of inflammatory cytokines and cytokine storms detected in COVID-19 cases. Low levels of protein S, due to pneumonia-induced VK depletion, were correlated with higher thrombogenicity and possibly fatal outcomes in COVID-19 patients.

Consuming a healthy diet is vital throughout the aging process to maintain and promote wellbeing. The aging population may be at risk for many suboptimal nutrient intakes, including VK, which have been shown to be associated with adverse health outcomes highly prevalent in this age group. Thus, the intake of VK-rich diets or VK supplements could prevent age-related diseases and/or support the effectiveness of medical treatments. However, more studies are needed to formulate the exactly recommended intakes of VK, including VK1, MK-4, and MK-7, due to their distinct bioavailability and biological activities. According to this review, higher values of VK intakes are needed, especially among the elderly and people who have comorbidities conditions that are most likely to be VK deficient.

## Figures and Tables

**Figure 1 antioxidants-10-00566-f001:**
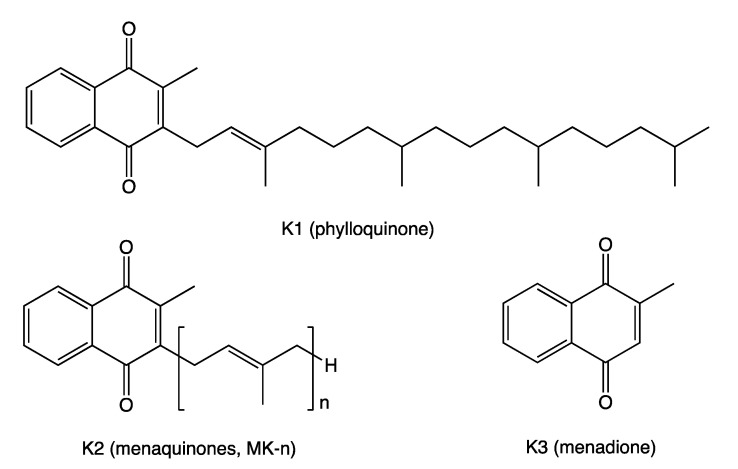
Vitamin K chemical structures. Vitamins K1 (VK1), K2 (VK2), and K3 (VK3) share the naphthoquinone ring; VK1 has a phytyl side-chain; VK2 has a side-chain with a varying number of isoprenyl units; VK3 has no side-chain.

**Table 1 antioxidants-10-00566-t001:** Vitamin K2: food category, sources, and amount.

Food Category	Food Source	VK2 *
Fermented foods	Natto Sauerkraut	850–1000 (90% MK-7, 8% MK-8) 5.5 (31% MK-6, 23% MK-9, 17% MK-5 and -8)
Hard cheeses		50–80 (15–67% MK-9, 6–22% MK-4, 6–22% MK-8)
Soft cheeses		30–60 (20–70% MK-9, 6–20% MK-4, 6–20% MK-8)
Eggs	Yolk	15–30 (MK-4)
Meats	Pork, beef, chicken	1.4–10 (MK-4)

*—μg/100 g food sample; MK-n—menaquinone.

**Table 2 antioxidants-10-00566-t002:** The effect of VK intake on bone outcome parameters.

Author, Year, Country [Ref.]	Subjects (W:M) Age (Mean ± SD)	Design (Length)	Intervention Exposure	Findings
Shiraki et al. 2000 Japan [44]	241 PMO 67.2 y	prospective 2 y	45 mg/d MK-4 vs. control	↓ ucOC (*p* < 0.0001) ↑ cOC (*p* = 0.0081) ↓ fracture risk (*p* = 0.0273)
Iwamoto et al. 2001 Japan [48]	72 PMO 65.3 y	prospective 2 y	45 mg/d MK-4 + Ca vs. Ca	↓ vertebral fractures (*p* < 0.0001) ↑ BMD (forearm) (*p* < 0.0001)
Purwosunu et al. 2006 Indonesia [49]	63 PMO 60.8 y	RCT 48 w	45 mg/d MK-4 + Ca vs. Ca	↓ ucOC (*p* ˂ 0.01) ↑ BMD (lumbar) (*p* < 0.05)
Bolton-Smith et al. 2007 UK [45]	244 healthy W 68.2 y	RCT 2 y	200 μg/d VK1 + 10 μg/d vitD3 + Ca vs. placebo	↓ ucOC (*p* < 0.001) ↑ BMD (ultradistal radius) (*p* < 0.01)
Knapen et al. 2007 Netherlands [50]	325 PMW 66.0 y	RCT 3 y	45 mg/d MK-4 vs. placebo	↑ BMC (*p* < 0.05) and bone strength (femoral neck)
Booth et al. 2008 USA [51]	452 (267:185) 68.4 y	RCT 3 y	500 μg/d PK vs. control	↓ ucOC (*p* ˂ 0.0001)
Cheung et al. 2008 Canada [52]	400 PMOa 59.1 y	RCT 2–4 y	5 mg/d VK1 vs. placebo	↓ fracture risk (*p* = 0.04)
Hirao et al. 2008 Japan [53]	44 PMW 68.4 y	prospective 1 y	45 mg/d VK2 + 5 mg/d alendronate vs. 5 mg/d alendronate	↓ ucOC (*p* = 0.014) ↓ ucOC:cOC (*p* = 0.007) ↑ BMD (femoral neck) (*p* = 0.03)
Tsugawa et al. 2008 Japan [54]	379 W 63.0 y	prospective 3 y	high VK1 vs. low VK1	↓ vertebral fracture risk (*p* < 0.001)
Binkley et al. 2009 USA [46]	381 PMW 62.5 y	RCT 1 y	1 mg/d VK1 or 45 mg/d MK-4 vs. placebo	↓ ucOC (*p* < 0.001) for both VK1 and MK-4 groups
Yamauchi et al. 2010 Japan [55]	221 healthy W 60.8 ± 9.5 y	cross-sectional	260±85 μg/d VK	↓ ucOC (*p* < 0.0001) ↑ BMD (lumbar) (*p* = 0.015)
Je et al. 2011 Korea [56]	78 PMW 67.8 y	RCT 6 mo	45 mg/d MK-4 + vitD + Ca vs. vitD + Ca	↓ ucOC (*p* = 0.008) ↑ BMD (lumbar) (*p* = 0.049)
Kanellakis et al. 2012 Greece [57]	173 PMW 62.0 y	RCT 12 mo	100 μg PK or MK-7 + vitD + Ca vs. control	↓ ucOC (*p* = 0.001) * ↑ BMD (lumbar) (*p* < 0.05) *
Knapen et al. 2013 Netherlands [58]	244 PMW 60.0 y	RCT 3 y	180 μg/d MK-7 vs. placebo	↓ ucOC (*p* < 0.001) ↑ BMD (lumbar spine, femoral neck), bone strength (*p* < 0.05)
Jiang et al. 2014 China [59]	213 PMW 64.4 y	RCT 1 y	45 mg/d MK-4 + Ca vs. Ca	↓ ucOC (*p* < 0.001) ↑ BMD (lumbar) (*p* < 0.001)
Rønn et al. 2016 Denmark [47]	148 PMOa 67.5 y	RCT 1 y	375 µg/d MK-7 vs. placebo	↓ ucOC (*p* < 0.05) ↓ ucOC:cOC (*p* < 0.05) ↑ bone structure (tibia) (*p* < 0.05)
Bultynck et al. 2020 UK [60]	62 (42:20) 80.0 ± 9.6 y	Prospective	↑ serum VK	↓ hip fracture risk
Moore et al. 2020 UK [61]	374 PMO 68.7 y	cross-sectional	↑ serum VK1	↓ fracture risk (*p* = 0.04)
Sim et al. 2020 Australia [62]	30 (10:20) 61.8 ± 9.9 y	RCT 12 w	136.7 μg/d VK	↓ ucOC and ucOC:tOC (*p* ≤ 0.01)

BMC—bone mineral content; BMD—bone mineral density; cOC—carboxylated osteocalcin; M—men; PMW—postmenopausal women; PMO—postmenopausal osteoporosis; PMOa—postmenopausal osteopenia; RCT—randomized controlled trial; SD—standard deviation; tOC—total osteocalcin; ucOC—undercarboxylated osteocalcin; W—women; ↑—increase; ↓—decrease. * for both VK1 and MK-4 groups.

**Table 3 antioxidants-10-00566-t003:** The effects of VK supplementation on vascular calcification.

Author, Year, Country (Ref.)	Subjects (W:M) Age (Mean ± SD)	Design (Length)	Intervention Exposure	Findings
Geleijnse et al. 2004 Netherlands [90]	4807 (2971:1836) 67.5 y	7 y	Q1 ˂ 21.6 μg/d VK2 Q2 21.6–32.7μg/d VK2 Q3 ˃ 32.7 μg/d VK2	↓ CHD mortality: RR = 0.43 (95% CI: 0.24–0.77, *p* = 0.005) Q3 vs. Q1 ↓ AC: OR = 0.48 (95% CI: 0.32–0.71, p ˂ 0.001) Q3 vs. Q1
Gast et al. 2009 Netherlands [91]	16,057 W 57.0 ± 6.0 y	Longitudinal 8.1 y	211.7μg/d VK1 29.1μg/d VK2	↓ CHD risk for 10 μg VK2: HR = 0.91 (95% CI: 0.85–1.00, *p* = 0.04)
Shea et al. 2009 USA [92]	388 (235:153) 68 y	RCT 3 y	500 μg/d VK1 vs. control	↓progression of CAC
Schurgers et al. 2010 France [93]	107 (43:64) 67 ± 13 y	18 mo	VK levels dp-ucMGP	↓ VK levels ↑ dp-ucMGP levels with CKD stage
Ueland et al. 2010 Norway [94]	147 (66:81) 74.0 ± 10 y	20 mo	VK levels dp-ucMGP	↓ VK levels ↑ dp-ucMGP in symptomatic AS
Schlieper et al. 2011 Serbia [95]	188 (89:99) 58 ± 15 y	Follow-up, 1104 days	VK levels dp-ucMGP dp-cMGP	↓ dp-cMGP ↑ CV: HR = 2.7 (95% CI: 1.2–6.2, *p* = 0.015) ↑ All-cause: HR = 2.16 (95% CI: 1.1–4.3, *p* = 0.027)
Ueland et al. 2011 Norway [96]	179 (39:140) 56 y	2.9 y	VK levels dp-ucMGP	↓ VK levels; ↑ dp-ucMGP ↑ heart failure: HR=5.62 (95% CI: 2.05–15.46, *p* = 0.001)
Westenfeld et al. 2011 Germany [97]	103 (48:55) ˃ 60.5 y	RCT 6 w	G1–45 µg/d MK-7 G2–135 µg/d MK-7 G3–360 µg/d MK-7	↓ dp-ucMGP by 77–93% G2 and G3 vs. control
Dalmeijer et al. 2012 Netherlands [98]	60 (36:24) 59.5 y	RCT 12 w	G1–180 μg/d MK-7 G2–360 μg/d MK-7	↓ dp-ucMGP by 31% G1 and 46% G2 vs. placebo
van den Heuvel et al. 2013 Netherlands [99]	577 (322:255) 59.9 ± 2.9 y	Follow-up 5.6 y	VK levels dp-ucMGP	↓ VK levels; ↑ dp-ucMGP ↑ CVD: HR=2.69 (95% CI: 1.09–6.62, *p* = 0.032)
Caluwé et al. 2014 Norway [100]	165 (83:82) 70.8 y	RCT 8 w	360, 720 or 1080 μg MK-7 thrice weekly	↓ dp-ucMGP by 17–33–46%
Liabeuf et al. 2014 France [101]	198 (40:158) 64 ± 8 y	Cross-sectional	VK levels dp-ucMGP	↓ VK levels; ↑ dp-ucMGP ↑ PAC: OR = 1.88 (95% CI: 1.14–3.11, *p* = 0.014)
Cheung et al. 2015 USA [102]	3401 (2245:1156) 61.9 y	Follow-up 13.3 y	↑ VK daily intake	↓ CVD mortality: HR = 0.78 (95% CI: 0.64–0.95, *p* = 0.016)
Knapen et al. 2015 Norway [103]	244 PMW 59.5 ± 3.3 y	RCT 3 y	180 µg/d MK-7 vs. placebo	↓ Stiffness Index β: −0.67 ± 2.78 vs. +0.15 ± 2.51, *p* = 0.018 ↓ cfPWV: −0.36 ± 1.48 m/s vs. +0.021 ± 1.22 m/s, *p* = 0.040
Kurnatowska et al. 2015 Poland [104]	42 (20:22) 58 y	RCT 270 days	90 μg/d MK-7 + 10 μg/d vitD vs. control	↑ CAC ↓dp-ucMGP
Asemi et al. 2016 Iran [105]	66 (31:35) 65.5 y	RCT 12 w	180 µg/d MK-7 + 10 µg/d vitD + 1 g/d Ca vs. placebo	↓ levels of left CIMT (*p* = 0.02) ↓ insulin (−0.9 vs. +2.6, *p* = 0.01) ↓ HOMA-IR (−0.4 vs. +0.7, *p* = 0.01)
Fulton et al. 2016 UK [106]	80 (36:44) 77 ± 5 y	RCT 6 mo	100 µg MK-7 vs. placebo	↓dp-ucMGP (p < 0.001)
Kurnatowska et al. 2016 Poland [107]	38 (17:21) 58.6 y	RCT 9 mo	90 μg/d MK-7 + 10 μg/d vitD vs. control	↓dp-ucMGP by 10.7%
Sardana et al. 2016 USA [108]	66 (6:60) T2D 62 ± 2 y	Cross-sectional	VK levels dp-ucMGP	↓ VK levels; ↑ dp-ucMGP ↑ cfPWV (*β* = 0.40, *p* = 0.011)
Aoun et al. 2017 Lebanon [109]	50 (20:30) 71.5 y	RCT 4 w	360 μg/d MK-7	↓ dp-ucMGP by 86%
Brandenburg et al. 2017 Germany [110]	99 (18:81) 69.1 y	RCT 1 y	2 mg/d VK1 vs. placebo	↓ progression of AVC (10.0% vs. 22.0%)
Shea et al. 2017 USA [111]	1061 (615:446) 74 ± 5 y	Follow-up 12.1 y	VK1 levels dp-ucMGP	↑ CVD risk in HBP patients (*n* = 489): HR = 2.94 (95% CI: 1.4–6.13, *p* ˂ 0.01)
Puzantian et al. 2018 USA [112]	137 (8:129) 59.6 y		VK levels dp-ucMGP	↓ VK levels; ↑ dp-ucMGP ↑ cfPWV (*β* = 0.21; *p* = 0.019)
Dal Canto et al. 2020 Netherlands [113]	601 (303:298) 70 ± 6 y	Follow-up 7 and 17 y	↓ VK levels ↓ vitD levels	↑ LVMI: *β* = 5.9 g/m^2.7^ (95% CI: 1.8–10.0 g/^2.7^) ↑ All-cause mortality: HR = 1.64 (95% CI: 1.12–2.39, *p* = 0.011)
Roumeliotis et al. 2020 Greece [114]	66 (31:35) diabetic CKD 68.5 ± 8.6 y	Follow-up 7 y	VK levels dp-ucMGP	↓ VK levels; ↑ dp-ucMGP ↑ CVD mortality: HR = 2.82 (95% CI: 1.07–7.49, *p* = 0.037)
Shea et al. 2020 USA [115]	3891 (2154:1737) 65 ± 11 y	Follow-up 13 y	↓ VK1 levels	↑ CVD risk: HR = 1.12 (95% CI, 0.94–1.33) ↑ All-cause mortality
Wessinger et al. 2020 USA [116]	60 (11:49) chronic stroke 61.7 ± 7.2 y	Cross-sectional	VK dietary intake	Among stroke survivors, 82% reported consuming below the Dietary Reference Intake for VK

AC—aortic calcification; AS—aortic stenosis; AVC—aortic valve calcification; CAC—coronary artery calcification; cfPWV—carotid-femoral pulse wave velocity; CHD—coronary heart disease; CIMT—carotid intima-media thickness; CKD—chronic kidney disease; dp-ucMGP —dephosphorylated—undercarboxylated matrix gla protein; CVD—cardiovascular diseases; HF—heart failure; HR—hazard ratio; LVMI—left ventricular mass index; M—men; MK—menaquinone; OR—odds ratio; PAC—peripheral arterial calcification; PMW—postmenopausal women; PMO—postmenopausal osteoporosis; PMOa—postmenopausal osteopenia; RCT—randomized controlled trial; RR—relative risk; SD—standard deviation; W—women; ↑—increase; ↓—decrease.

**Table 4 antioxidants-10-00566-t004:** The effect of VK intake on metabolic disorders.

Author, Year, Country [Ref.]	Subjects (W:M) Age (Mean ± SD)	Design (Length)	Intervention Investigations	Findings
Im et al. 2008 South Korea [129]	339 PMW T2D 57.2 y		Biochemical and hormonal parameters for (1) NG; (2) IGF; (3) T2D groups	↓ OC in (3) vs. (1) (*p* < 0.005) OC levels—inversely correlated with FG (*r* = −0.195, p < 0.001), HbA1c (*r* = −0.219, p < 0.001), FI (*r* = −0.131, *p* < 0.016), HOMA-IR (*r* = −0.163, *p* < 0.003)
Yoshida et al. 2008 USA [130]	355 (213:142) 68 y	RCT 36 mo	500 μg/d PK vs. control	↓ HOMA-IR (*p*-adjusted < 0.01) and ↓ plasma insulin (*p*-adjusted < 0.04)—only for men ↓% ucOC (*p* < 0.001) for both men and women
Kanazawa et al. 2009 Japan [131]	329 (149:179) 65.8 y		Biochemical and hormonal parameters	Negative correlation between OC and FG and HbA1c (for all: *p* < 0.05),% fat, baPWV and IMT in men (*p* < 0.05) Positive correlation between OC and total adiponectin in PMF (*p* < 0.001)
Kindblom et al. 2009 Sweden [132]	1010 M 857 non-T2D 153 T2D 75.3 ± 3.2 y	MrOS Sweden study	Biochemical and hormonal parameters	↓ OC in T2D (−21.7%, *p* < 0.001) vs. non-T2D Plasma OC—inversely correlated with BMI, fat mass, and plasma glucose (*p* < 0.001)
Shea et al. 2009 USA [133]	348 (206:142) non-T2D 68 y	Cross sectional 3 y	OC levels (tOC, ucOC, cOC) and HOMA-IR	↑ cOC and tOC were associated with ↓ HOMA-IR (*p* = 0.006 and *p* = 0.02, respectively)
Bao et al. 2011 China [134]	181 M 76 non-metS 105 metS 64.9 ± 10.7 y		Biochemical and hormonal parameters	↓ OC in MetS vs. non-MetS (*p* < 0.001); OC was independently associated with metS (OR = 0.060, 95% CI: 0.005–0.651)
Alfadda et al. 2013 Saudi Arabia [135]	203 T2D ± MetS 52.5 ± 9.6 y	Cross-sectional	Biochemical and hormonal parameters	↓ tOC (*p* = 0.01) and ucOC (*p* = 0.03) in metS vs. non-metS. Positive correlation between ucOC and HDL-C (*p* = 0.023). Negative correlation between tOC and HbA1c (*p* = 0.01) and serum TGs (*p* = 0.049.
Confraveux et al. 2014 France [136]	798 M 65.3 ± 7 y	MINOS study	Biochemical and hormonal parameters	Negative correlation between OC and glycemia (*p* < 0.0001)
Shea et al. 2017 USA [137]	401 (237:164) 69 ± 6 y	RCT 3 y	500 μg/d PK (+Ca and vitD) vs. control (Ca and vitD)	↓ ucOC (*p* < 0.001)
Knapen et al. 2018 Netherlands [138]	214 PMW 60 y	RCT 3 y	180 µg/d MK-7 vs. placebo	↑ cOC (*p* < 0.0001) ↓ ucOC (*p* < 0.0001)
Dumitru et al. 2019 Romania [139]	146 PMW T2D 62.1 y	Cross sectional 30 mo	Biochemical and hormonal parameters in T2D group vs. control	↓ tOC (*p* < 0.05) in T2D group Negative correlation between tOC and HbA1c, BMI, TGs (for all: *p* < 0.05), and HDL-C (*p* = 0.001)
Guney et al. 2019 Turkey [140]	191 PMW metS 56 y	cross-sectional	Biochemical and hormonal parameters in metS group vs. control	↓ OC (*p* < 0.001) in metS group Positive correlation between vitD and OC (*r* = 0.198; *p* = 0.008) Negative correlation between OC and hs-CRP (*p* = 0.003), HOMA-IR (*p* = 0.048), and HbA1c (*p* = 0.001)
Aguayo-Ruiz et al. 2020 Mexico [141]	40 (24:16) T2D 56 y	RCT 3 mo	(1) 100 µg/d K2 (2) 100 µg/d K2+vit D3 (3) vit D3	(1): ↓ glycemia (*p* = 0.002) ↑ cOC (*p* < 0.041) (2): ↓ glycemia (*p* = 0.002)
Jeannin et al. 2020 France [142]	198 (40:158) T2D 64 ± 8.4 y	Cohort	NDS, dp-ucMGP in plasma	↑ peripheral NDS (15.7%) correlated with dp-ucMGP (*r* = 0.51, *p* < 0.0001)
Sakak et al. 2020 Iran [143]	68 (42:26) T2D 57.6 y	RCT 12 w	360 μg MK-7 vs. placebo	↓ FPG (*p*-adjusted = 0.031) ↓ HbA1c (*p*-adjusted = 0.004) ↓ HOMA-IR (*p* = 0.019) vs. baseline

BMI—body mass index; cOC—carboxylated osteocalcin; dp-ucMGP—dephospho-uncarboxylated matrix-gla-protein; FG—fasting glucose; FI—fasting insulin; FPG—fasting plasma glucose; FPβC—functional pancreatic β cells; HbA1c—glycosylated hemoglobin; HDL-C—high—density lipoprotein cholesterol; HOMA-IR—homeostatic model assessment of insulin resistance; hs-CRP—high sensitive C—reactive protein; IFG—impaired fasting glucose; M—men; metS—metabolic syndrome; NDS—neuropathy disability score; NG—normal glucose; OC—total carboxylated osteocalcin; PMO—postmenopausal osteoporosis; PMW—postmenopausal women; RCT—randomized controlled trial; SD—standard deviation; tOC—total osteocalcin; ucOC—undercarboxylated osteocalcin; T2D—type 2 diabetes; TGs—triglycerides; W—women; ↑—increase; ↓—decrease.

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
