# Peer review of "The Role of Vitamin K in Humans: Implication in Aging and Age-Associated Diseases"

_antioxidants, 2021, doi:10.3390/antiox10040566_

Round 1

Reviewer 1 Report

The number of reviews focused on the implication of the role of vitamin K in aging is apparently scarce. Thus, the authors cover an interesting topic given the serious health problems associated with such condition state. The paper submitted by the authors is well built and structured with properly sections treated in an exhaustive way. In the field of (clinical)nutrition is difficult to associate health benefits to only individual factors, a synergic and complementary effect being involved on numerous occasions. The connection of vitamin K with other ones and with mineral substances is traced when necessary emphasizing de role of rich antioxidants food (e.g. fruit and vegetables) in the diet. I must admit that the paper is so dense that sometimes its reading fatigue, but this is undoubtedly a necessary evil, for the sake of the rigor and completeness with which the subject is approached. My only doubt is whether the inclusion of the Tables is necessary or can instead be sent as supplementary material, thus saving space in the text. The writing of the text is original; the match percentage excluding references as checked by the “turniting” program is almost nil. In short, it is dealt a serious and fine work, a nice paper, and in my opinion, it should be published in ANTIOXIDANTS.

Author Response

Thank you for your comments. Our review has one figure and four tables which underline the aim of the study, and with all due respect we consider keeping them in the main text.

Reviewer 2 Report

This manuscript, in general is well-written and present a compendium on the properties of vitamin K and its role in the pathogenesis of some human disorders. I cannot see serious flaws in this manuscript, but have some minor remarks.

Somme grammar issues should be addressed.

I am not sure, whether the term “mature adults” is right. Maybe older adults?

The titles of section should be unified in style – eg., if section 2. is entitled Vitamin in Bone Health, so section 3 should be also entitled in similar fashion, eg., Vitamin D in the prevention and therapy of Vascular Calcification and Cardiovascular Diseases

Introduction

“Oxidative stress (OS) and chronic inflammation are fundamental patho-physiological mechanisms in the aging progression” – oxidative stress and chronic inflammation should not be presented as independent phenomena.

Paragraph lines 83-88 must be rewritten as it is confusing. If, as authors state, vitamin K directly uptakes ROS, so it should exert general antioxidant effect if in a suitable concentration, and not only against lipid peroxidation. Furthermore, if so – “blocking free radical accumulation” is obvious. The last sentence in that paragraph is redundant.

Lines 125-126 – references are needed.

The section 4 title is somehow misleading as it suggests a positive association.

Section 5 – please consider a change of title and including some information on age-related macular degeneration (AMD).

Paragraph lines 634-640 should be rewritten as it is somehow chaotic, non-cohesive, too general and requires references.

Line 650 Cancer cell death phenotype induced -> Cancer cell death induced.

Section 6 – the authors should conclude whether vitamin K has a pro- or anti-autophagic effect in cancer and whether autophagy-related mechanism of vitamin K is pro-death or pro-life. In the present form it is unclear.

The concluding section might be more comprehensive.
